# New RT-PCR Assay for the Detection of Current and Future SARS-CoV-2 Variants

**DOI:** 10.3390/v15010206

**Published:** 2023-01-11

**Authors:** Antonio Marchini, Mauro Petrillo, Amy Parrish, Gerhard Buttinger, Simona Tavazzi, Maddalena Querci, Fay Betsou, Goffe Elsinga, Gertjan Medema, Tamir Abdelrahman, Bernd Gawlik, Philippe Corbisier

**Affiliations:** 1European Commission, Joint Research Centre (JRC), 2440 Geel, Belgium; 2Seidor Italy S.r.l., 20129 Milan, Italy; 3Department of Microbiology, Laboratoire National de Santé, 3583 Dudelange, Luxembourg; 4European Commission, Joint Research Centre (JRC), 21027 Ispra, Italy; 5Biological Resource Center of Institut Pasteur, Université Paris Cité, 75015 Paris, France; 6KWR Water Research Institute, 3433 PE Nieuwegein, The Netherlands

**Keywords:** SARS-CoV-2, SARS-CoV-2 ultra-conserved genomic elements, RT-PCR diagnostic assays, WHO recommended assays, wastewater surveillance

## Abstract

Multiple lineages of SARS-CoV-2 have been identified featuring distinct sets of genetic changes that confer to the virus higher transmissibility and ability to evade existing immunity. The continuous evolution of SARS-CoV-2 may pose challenges for current treatment options and diagnostic tools. In this study, we have first evaluated the performance of the 14 WHO-recommended real-time reverse transcription (RT)-PCR assays currently in use for the detection of SARS-CoV-2 and found that only one assay has reduced performance against Omicron. We then developed a new duplex real-time RT-PCR assay based on the amplification of two ultra-conserved elements present within the SARS-CoV-2 genome. The new duplex assay successfully detects all of the tested SARS-CoV-2 variants of concern (including Omicron sub-lineages BA.4 and BA.5) from both clinical and wastewater samples with high sensitivity and specificity. The assay also functions as a one-step droplet digital RT-PCR assay. This new assay, in addition to clinical testing, could be adopted in surveillance programs for the routine monitoring of SARS-CoV-2’s presence in a population in wastewater samples. Positive results with our assay in conjunction with negative results from an Omicron-specific assay may provide timely indication of the emergence of a novel SARS-CoV-2 variant in a certain community and thereby aid public health interventions.

## 1. Introduction

COVID-19, caused by the severe acute respiratory syndrome coronavirus 2 (SARS-CoV-2), is one of most deadly pandemics in human history with over 605 million confirmed cases reported worldwide and over 6.5 million deaths at the time of writing. This pandemic has been characterized by repeated waves of transmission triggered by new variants with increased fitness, which in turn replace existing variants. Multiple variants of SARS-CoV-2 have been described, of which a few are considered variants of concern (VOCs), given their impact on public health. For definition, VOCs are characterized by mutations in the viral genome that (i) increase transmissibility or detrimentally change COVID-19 epidemiology, or (ii) increase virulence or change clinical disease presentation, or (iii) decrease the effectiveness of public health and social measures or the efficacy of available diagnostics, vaccines, and therapeutics. Based on the recent epidemiological update by the World Health Organization (WHO), as of 11 July 2022, five SARS-CoV-2 VOCs have been described since the beginning of the pandemic: Alpha (B1.1.7), Beta (B.1.351); Gamma (P1), Delta (B.1.617.2), and Omicron (B.1.1.529) [1,2].

In addition to VOCs, eight variants of interest (VOIs) have been described by the WHO, namely Epsilon (B.1.427 and B.1.429), Zeta (P.2), Eta (B.1.525), Theta (P.3), Iota (B.1.526), Kappa (B.1.617.1), Lambda (C.37), and Mu (B.1.621) [2]. VOIs feature mutations in their viral genomes that are proven or predicted to affect virus characteristics such as transmissibility, disease severity, immune escape, diagnostic or therapeutic escape, and the ability to evade existing immunity. Although these variants caused local outbreaks, more transmissible VOCs outnumbered and replaced them.

Omicron, and its descent lineages, is currently the dominant variant circulating globally, responsible for almost all new COVID-19 cases. Omicron, which was first reported in late November 2021 [3], has accumulated about 60 mutations, in comparison to the Wuhan H-1 original strain, including 30 amino acid changes within the S gene which encodes the spike protein [4,5]. The spike protein is involved in host–receptor engagement and viral entry and is the main target of neutralizing antibodies developed against the virus [6,7]. Owing to these mutations, Omicron features higher rates of transmission and an increase in immune evasion [8,9]. In recent months, there has been a trend for second-generation descendants of Omicron namely BA.1, BA.2, BA.3, BA.4, and BA.5 with the proportion of BA.5 continuing to increase in August–September 2022. Growing evidence indicates that these mutations are associated with a decrease in therapeutics or vaccination effectiveness [10,11,12,13].

It is not certain where and/or when a new variant to replace the Omicron variant will emerge, but there is a common scientific assumption that SARS-CoV-2 will continue to evolve. Millions of virus genome sequences have also revealed that not all of the viral genome is subjected to the same mutational rate [14]. There are “conserved” elements within the viral genome that do not tolerate changes or mutate with very low frequencies, suggesting that their integrity is crucial for viral fitness [15,16].

Testing plays a central role in the response to the COVID-19 pandemic. The emergence of novel and evolving variants of SARS-CoV-2 has fostered the need for updated diagnostic methods for the detection of SARS-CoV-2 infections. The gold standard for the detection of SARS-CoV-2 is the real-time reverse transcription (RT)-PCR. Multiple assays have been developed for the specific detection of SARS-CoV-2 (reviewed in [17]), and some of these assays (e.g., WHO-recommended assays) are commonly used by laboratories. RT-PCR technology relies on the amplification of specific viral RNA genomic sequences. Commonly in use, SARS-CoV-2 assays target sequences present within the ORF1ab, RNA-dependent RNA polymerase (RdRp)/helicase, envelope (E), nucleocapsid (N), and spike (S) genes.

Viral mutations can potentially affect the sensitivity and specificity of a certain method, leading to alterations in test performance characteristics, including false negatives [18]. To circumvent these problems, multi-target assays have been developed reducing the risk that some SARS-CoV-2 infections may evade detection [18,19,20,21].

Several studies have compared the performance of RT-PCR assays for SARS-CoV-2 detection [22,23,24]. The results have shown that generally the analytical sensitivity and specificity of the different assays is high and quite comparable between the assays. However, some decreases in performance have also been reported especially for those assays that target the *N* and *S* genes [23]. This correlates with the high mutational rate of these two genes that may result in mismatches between the oligonucleotides of the assays and their consensus sequences. Continued evaluation of the performance of established tests together with the development of novel updated and reliable tests should go hand-in-hand with the evolution of SARS-CoV-2. We have recently described a one-step RT-PCR assay for the specific detection of Omicron [25]. In contrast to other RT-PCR assays based on the deletion H69–/V70, we have proven that this assay was also able to detect Omicron BA.1 and BA.2 and the currently dominant BA.4 and BA.5 lineages. In this study, we first evaluated in silico the matchings of the primers and probes of the WHO-recommended RT-PCR-based assays against the consensus SARS-CoV-2 Omicron genome sequences deposited in GISAID [26] and predicted their performance. We then established a novel and “universal” RT-PCR method that specifically targets ultra-conserved elements (UCEs) of the viral genome. Based on our in silico analysis carried out by aligning all of the SARS-CoV-2 consensus sequences, these UCEs have not mutated since the appearance of SARS-CoV-2. This approach can detect currently circulating variants and we anticipate that it could also detect new variants that will eventually emerge as the selected UCEs are not predicted to mutate in the future.

## 2. Materials and Methods

### 2.1. Generation of Lineage Consensus Sequences

Lineages with at least 10 assigned SARS-CoV-2 sequenced genomes were considered in this analysis. Each consensus sequence is representative of all the known SARS-CoV-2 sequences deposited in GISAID [26] with a certain set of mutations. We considered as characteristic mutations for each lineage, all mutations that are found in at least 90% of sequences assigned to that lineage. An in-house developed script was used to recover lineage mutations from the Broad Institute COVID CG [27] application programming interface (API) (https://covidcg.org/group_mutation_frequencies, accessed on 5 October 2022) invoked with the following parameters: group = lineage, mutation type = dna, and consensus_threshold = 0.9.

The lineages’ consensus sequences were generated by an in-house developed script using the SARS-CoV-2 isolate Wuhan-Hu-1 complete genome (NCBI Reference Sequence: NC_045512.2) as a reference sequence and in silico introducing the retrieved mutations characteristic of each lineage. In this way, 1815 consensus sequences were generated, which together are representative of a total of 10,452,789 GISAID sequence submissions.

### 2.2. Identification of SARS-CoV-2 Ultra-Conserved Elements and Design of RT-PCR Assays

An in-house developed script was used to identify SARS-CoV-2 ultra-conserved elements (UCEs). All consensus sequences were used to create a SARS-CoV-2 sequence containing all known observed genomic mutations (masked with N). Based on this sequence, fragments of 500 nucleotides harboring fewer than 10 mutations were identified. These fragments were manually selected and fed into Primer3Plus [28,29] to pick up primers and probes suitable for qPCR with “qPCR” as the load server settings. Eleven potential qPCR methods were designed, and all of them were tested by running in silico PCR simulations (as described in Section 2.3) using the consensus sequences (generated as described in Section 2.1) as a template. Only methods which were able to detect at least 95% of the consensus sequences were considered for further analyses. Among the best ranking five, the first two designed methods, called JRC-CoV-UCE.1 and JRC-CoV-UCE.2 here, were selected for further validation of the clinical samples.

### 2.3. In Silico PCR Simulations

All of the in silico PCR simulations were run by using the thermonucleotideBLAST software version v.2.17 [30], installed locally. Default parameters were used, with the exception of the following: minimum primer Tm (−e) = 20, minimum probe Tm (−E) = 20, and maximum amplicon length (−l) = 200. Outputs of thermonucleotideBLAST runs were parsed by an in-house developed script to make the analyses reported in the paper, i.e., the number of mutations on oligonucleotides target regions.

### 2.4. Samples, Sample Processing, and RNA Extraction Procedures

Clinical samples. Anonymized RNA extracts from positive SARS-CoV-2 nasopharyngeal (NP) swab samples were obtained as per request from the LuxMicroBiobank at the Laboratoire National de Santé (LNS), Dudelange, Luxembourg. Following the original positive diagnosis by RT-PCR with the CE-IVD Allplex SARS-CoV-2 Assay (Seegene Inc., Seoul, South Korea), the clinical NP samples underwent nucleic acid extraction using the Starlet Instrument (Seegene Inc., Seoul, South Korea) with the STARMag Universal Cartridge Kit (Seegene Inc., Seoul, South Korea). Prior to use, positive sample RNA extracts were stored at −80 °C in the LuxMicroBiobank at the LNS within the capacity of the National Reference Laboratory for acute respiratory infections and in the frame of the national surveillance program for SARS-CoV-2. The clinical samples that were used throughout this paper were sequenced using Illumina technologies and assigned to a pangolin variant classification using PangoLearn v.3.1.5 (accession date 21 July 2022) [31]. All sequencing data from Luxembourg are publicly available on GISAID and provided in the Appendix A.

Wastewater samples. Wastewater samples were transported to the laboratory on melting ice and RNA was isolated within 24 h after sampling from a volume of 50 mL. Larger particles (debris and bacteria) were removed from the samples by pelleting using centrifugation of the sample in 50 mL conical centrifuge tubes at 4654× *g* for 30 min without brake. The supernatant was filtered through Centricon^®^ Plus-70 centrifugal ultrafilters with a cut-off of 30 kDa (Millipore, Amsterdam, The Netherlands) by centrifugation at 3000× *g* for 20 min. The Centricon concentrate was between 0.44 and 1.79 g. RNA was extracted from Centricon wastewater concentrates by magnetic bead-based extraction using the Biomerieux Nuclisens EasyQ^®^ kit (Biomerieux, Amersfoort, The Netherlands) in combination with the semi-automated KingFisher mL (Thermo Scientific, Bleiswijk, The Netherlands) purification system as previously described [32,33].

Viral genomes. Synthetic full-length RNA viral genomes of the SARS-CoV-2 Wuhan Hu-1 lineage and indicated variants were purchased from Twist Bioscience (South San Francisco, CA, USA) and diluted 10 time in nuclease free-water supplemented with carrier RNA, before use at a calculated concentration of 5 × 10^5^ cp/RT-PCR reaction.

### 2.5. RT-PCR Testing

The sequences of the primers and TaqMan^®^ probes for the JRC-CoV-UCE.1 and JRC-CoV-UCE.2 assay are provided in Table 1 together with the amplified SARS-CoV-2 regions.

The TaqMan^®^ probes for the JRC-CoV-UCE.1 and JRC-CoV-UCE.2 assays were labelled at the 5′-end with the reporter molecules VIC and ABY™, respectively (Applied Biosystems, Waltham, MA, USA). Both probes were labelled at the 3′-end with the proprietary QSY ™ succinimidyl ester quencher (QSY). JRC-CoV-UCE.1 targets a region of 89 nucleotides (nt) located in the *Orf1a(NSP3)* gene corresponding to nt 4595–4683 of the SARS-CoV-2 genome (sequence NC_045512.2), while the JRC-CoV-UCE.2 targets a region of 85 nt located in the *Orf1a(NSP3)* corresponding to nt 12658–12742. JRC-CoV-UCE.1 and JRC-CoV-UCE.2 were both optimized to run in a duplex assay. The reaction mixtures for the RT-PCR experiments were prepared as follows: for each 25 µL reaction, 5 µL of the extracted RNA was added to 20 µL of the reaction mix containing 7.75 µL of H_2_0 (RNAse free), 1 µL/each of the forward and reverse primers, (at a final concentration of 900 nM),1 µL/each oflabelled probes (at a final concentration of 400 nM) and 6.25 µL of MasterMix (TaqPath 1-step RT-qPCR MasterMix 4 x, ThermoFisher Scientific). The PCR program consists of uracil–DNA glycosylase incubation (2 min at 25 °C), reverse transcription (15 min at 50 °C), TaqMan^®^ activation (2 min at 95 °C) and 45 cycles of amplification (3 s at 95 °C followed by 30 s at 64.5 °C). When the synthetic RNA viral genomes were analyzed, an annealing temperature of 62 °C was chosen. All of the RT-PCR reactions were performed in a calibrated QuantStudio 7 Flex Real-Time PCR System (Applied Biosystems, Waltham, Massachusetts, United States); the raw data were analyzed with the QuantStudio software (version 1.3) with an automatic threshold.

For clinical samples, a control run was performed in tandem using the AllPlex SARS-CoV-2 Assay (Seegene, Seoul, South Korea), which can simultaneously detect four target genes (*RdRP/S* and *N* and *E* genes) in one reaction, according to the manufacturer’s instructions. This kit is used in routine diagnosis at the LNS. PCR experiments were run on a BioRad CFX96 Real-Time PCR System. For the 40 Omicron-positive clinical samples, the same samples were also analyzed a second time at the JRC-Geel using a calibrated QuantStudio 7 Flex Real-Time PCR System (Applied Biosystems, Waltham, MA, USA).

For the wastewater samples, the 2019-nCoV N2 [34,35] and E Sarbeco [36] methods were used as reference methods according to the published protocols. Each individual reaction contained 5 μL of the total volume of 100 μL eluted RNA template. Four μL 5× Taqman Fast Virus 1-Step Master Mix (Applied Biosystems, Fisher Scientific, Landsmeer, The Netherlands) and 2 μL of 4 mg/mL BSA (Bovine Serum Albumin, Roche Diagnostics, Almere, The Netherlands) were used for each individual reaction on a total volume of 20 μL. Two hundred nM of primers and probes were used for the 2019-nCoV N2 assay; 400 nM of primers and 200 nM of the probe were used for the E Sarbeco assay. Thermal cycling reactions were carried out at 50 °C for 5 min, followed by 45 cycles of 95 °C for 10 s, 55 °C for 10 s, and 60 °C for 20 s on a CFX96 Touch Real-Time PCR Detection System (Bio-Rad Laboratories, Veenendaal, The Netherlands). The reactions were considered positive if the cycle threshold was below 40 cycles. In addition, the samples were also analyzed by the specific Omicron OmMet assay [25].

### 2.6. Annealing Temperature Optimization

To determine the optimal annealing temperature, six Omicron-positive clinical samples of varying viral load were used. Each sample was subjected to eight different annealing temperatures in individual wells, ranging from 55.0 to 70.0 °C, with no other modifications to the JRC-CoV-UCE protocol. An optimal annealing temperature of 64.5 °C was chosen by determining the difference from the average Ct value of JRC-CoV-UCE.1 and JRC-CoV-UCE UCE.2 from the Ct value obtained with the Seegene AllPlex SARS-CoV-2 Assay, taking into account relative standard deviations.

### 2.7. Quantification of Viral Load by Droplet Digital RT-PCR (ddRT-PCR)

To measure the viral load present in the clinical sample used to determine the limit of detection (LoD) of the assays listed in Table 2 (see below), JRC-CoV-UCE.1 and JRC-CoV-UCE.2 were run as a duplex ddRT-PCR assay maintaining the same annealing temperatures, primers, and probe concentrations described for the RT-PCR assays (Table 1). As a confirmatory assay, the CDC-N (CDC, Beijing, China) assay [37] was also used (the primers and annealing temperature for this assay are reported in Table 2). The one-step RT-ddPCR Advanced Kit for Probes (Bio-Rad Laboratories, Hercules, CA, USA; Cat. Nr. 186-4021) was used according to the manufacturer’s recommendations. A clinical sample containing Omicron (BA.1) was properly diluted. Each 20 µL ddRT-PCR reaction consisted of 5 µL of the diluted sample, 1 µL 300 mM DTT, 5 µL Supermix, 2 µL reverse transcriptase, 0.8 µL of each primer, 0.8 µL (UCE assays), or 0.4 µL (CDC-N) of probe and RNAse-free water up to 20 µL. Droplets were generated with the AutoDG*^®^* and read with a QX200 reader (Bio-Rad Laboratories, Hercules, CA, USA). RT incubation was performed at 50 °C for 60 min followed by an enzyme activation step at 95 °C for 10 min. Forty-five amplification cycles were run with 30 *s* at 95 °C and 60 s at the assays’ annealing temperature. Thereafter the enzyme was deactivated at 98 °C for 10 min. The plate was kept at 4 °C until reading. The read-out of droplets with positive and negative signals was performed with a manual fluorescence amplitude threshold setting with the combined wells option using the QuantaSoft software (version 1.7.4.0917) and the Bio-Rad droplet readers QX200 to observe a clear segregation between the positive and negative droplets. The target RNA concentration was calculated from the fraction of positive droplets (positive end-point reactions) and the number of accepted droplets using Poisson statistics taking into account the dilution factors and droplet volumes of 0.735 nL for the Supermix.

### 2.8. Limit of Detection Calculation

The LoDs of JRC-CoV-UCE.1, JRC-CoV-UCE.2, and of the RT-PCR assays listed in Table 2 were determined using a clinical sample as a template with a defined -SARS-CoV-2 copy number (calculated by ddPCR as described in Section 2.7). JRC-CoV-UCE.1 and JRC-CoV-UCE.2 were run together as a duplex assay. A 1:10 serial dilution (D1:D6) of the clinical sample was prepared and analyzed by the different assays. The sequences of the primers and TaqMan*^®^* probes and the PCR conditions of the various assays are provided in Table 1 and Table 2.

## 3. Results

### 3.1. Omicron Mutations Affect Only One of the Real-Time RT-PCR Assays Recommended by WHO for the Detection of SARS-CoV-2

To date, the scientific community has sequenced more than 13 million SARS-CoV-2 genomes and the results have been shared worldwide [26]. Thanks to these efforts, we grouped most of the GISAID-submitted sequences by generating 1815 different consensus sequences of which 203 are Omicron-related (the numbers include 28 chimeric lineages). Each consensus sequence groups all of the whole genome sequences with the same set of mutations and together they represent a total of 10,452,789 sequence submissions. Omicron has accumulated over 60 genetic mutations in comparison with its Wuhan-H1 prototype, which appeared for the first time in late December 2019 [2]. These mutations pose the question as to whether the current methods, which were developed at the beginning or during the first year of the pandemic against other VOCs, would still be able to detect Omicron lineages with similar efficiency. We focused on the 14 RT-PCR-based methods recommended by the WHO as these are among the most widely used in clinical testing (https://www.who.int/docs/default-source/coronaviruse/whoinhouseassays.pdf, accessed on 5 October 2022) (Appendix A).

We carried out in silico alignment of primers and probes sequences used in the various assays against the 1815 full-length SARS-CoV-2 genomic consensus sequences. A number of mismatches were found in the primers/probes of all of the assays when aligned with their corresponding target sequences (Appendix A). For five assays, mismatches were found in a large proportion of the sequences analyzed (Appendix A). Despite this, the number of mismatches for single consensus sequence was limited in most of the cases to only one or two, suggesting that these mutations would not reduce PCR efficiency. Some examples of mismatches are illustrated in Figure 1. The complete list of mismatches for all of the assays is presented in Appendix A.

In particular, we found that the primer sets for the 2019-nCoV N2 (CDC, US) [34,35], CDC ORF1ab (China CDC) [37], HKU-ORF1b-nsp14 [38], and N (HKU, Hong Kong SAR) assays, although developed before the appearance of Omicron, still perfectly align with their Omicron genome consensus sequences (Appendix A). Specific alignment with the subgroup of the BA.4 and BA.5 sequences shows a perfect match for additional assays, namely NIID 2019 nCOV N (NIID, JP) [39], RdRp gene/nCoV_IP2 and RdRp gene/nCoV_IP4, (IP, France) [40], and HKU-N (HKU, HK) [38]. Similar results were also obtained by analyzing the consensus sequences of the most recent BA.2.75, BQ.1, and XBB Omicron subvariants (Appendix A). An exception to this general trend, was the WH-NIH assay (NIH, Nonthaburi, Thailand) [41] which presents a ninenucleotide mismatch due to a deletion in the consensus sequence of its reverse primer (Figure 1).

We performed in silico PCR simulations using these assays against Omicron consensus sequences. The analysis predicted that most of the assays would continue to efficiently detect almost 100% of these sequences. On the other hand, the WH-NICN assay (NIH, Nonthaburi, Thailand) was the only assay predicted to fail in detecting most of the Omicron sequences and three other assays, namely 2019-nCoV N1, E Sarbeco, and CDC-N were predicted to have reduced sensitivity (Appendix A).

We checked the performance of the assays featuring the highest number of mismatches in their oligonucleotides. Synthetic, full-length, single-stranded RNA genomes of the Wuhan H1 prototype or the Omicron (BA.1) variant were used as templates for the RT-PCRs. The WH-NIC N assay showed a reduced ability to detect Omicron RNA in comparison with the Wuhan H1 RNA (approximately 8 Ct of difference using the same input amounts of synthetic RNA viral genomes) (Figure 2). On the contrary, the 2019 nCoV N1, CDC-N, and E Sarbeco assays, despite presenting mismatches in their primers against the corresponding Omicron consensus sequence, recognized both Omicron and Wuhan H1 sequences with similar efficiency. In conclusion, our results indicate that the WHO-recommended assays with the exception of WH-NIC N remain as very reliable assays against the Omicron variant.

### 3.2. Development of a Novel Duplex Real-Time (dd)RT-PCR Assay Targeting the Ultra-Conserved Elements of the SARS-CoV-2 Genome

Although our analysis provides evidence that, with the exception of the WH-NIC-N, the WHO-recommended assays are still reliable for routine COVID-19 diagnosis, it is still possible that acquisition of further mutations within their primer sequences may decrease their sensitivity. To minimize the impact of viral mutations for the monitoring of SARS-CoV-2, we decided to take advantage of the enormous amount of data from publicly available genome sequencing in order to design a new RT-PCR assay based on the specific amplification of the ultra-conserved elements present in the SARS-CoV-2 genome. We anticipated that such an assay would be able to detect not only all of the SARS-CoV-2 variants that are presently circulating globally, but also those that will presumably emerge in the future. By aligning all of the SARS-CoV-2 consensus sequences, we searched for genomic regions that were conserved or with very low frequencies of mutations (here they are called SARS-CoV-2 ultra-conserved elements, UCEs), from which potential primer pairs could be designed. We selected two UCEs, that would particularly fit to our purpose, UCE.1 and UCE.2, which are located in the Orf1a and in particular within the open reading frames coding non-structural proteins 3 (NSP3) and 9 (NSP9), respectively (Figure 3A). We designed primer sets and probes that would specifically amplify the conserved regions and we named the two individual assays as JRC-CoV-UCE.1 and JRC-CoV-UCE.2 (Table 1), and when run as a unique duplex assay, this was termed JRC-CoV-UCE.

For each individual assay, we first performed an in silico PCR simulation analyses on a subset (about 10%) of complete high-quality SARS-CoV-2 genomic sequences randomly retrieved from GISAID and on the 1815 consensus sequences of all the variants. JRC-CoV-UCE.1 and JRC-CoV-UCE.2 were predicted to recognize most of the SARS-CoV-2 sequences with less than 0.02% of the total sequences analyzed presenting mismatches with the JRC-CoV-UCE oligonucleotides (Figure 3B and Appendix A). In particular, only 398 out of 4,266,204 Omicron sequences presented a single mismatch in the primer or probe of JRC.CoV.UCE.1 while no mismatches were found for JRC CoV.UCE.2).

We then tested the duplex JRC-CoV-UCE method on samples containing 5 × 10^5^ copies of synthetic full-length RNA genomes of the isolate Wuhan Hu-1 or of the Alpha (B.1.1.7), Beta (B.1.351), Gamma (P.1), Delta (B.1.617.2), Epsilon (B.1.429), Iota (B.1.526), Kappa (B.1.617.1), and Omicron (BA.1) variants. JRC-CoV-UCEs detected all of the viral genomes with similar efficiency, reflecting the high degree of conservation of the amplified regions (Figure 4A).

We then carried out a temperature gradient optimization experiment using six clinical samples containing Omicron to find the optimal annealing temperature for the method. In these experiments, the JRC-CoV-UCE duplex was run at different annealing temperatures and compared with the control Seegene AllPlex SARS-CoV-2 Assay, run at its optimal annealing temperature of 60 °C. This assay is widely used and known to be a quite robust and reliable assay for the detection of SARS-CoV-2 [42,43].

JRC-CoV-UCE performed with similar efficiency in the 60–64.5 °C temperature range. At an annealing temperature of 64.5 °C, the performance of JRC-CoV-UCE was slightly superior to that of the Seegene AllPlex SARS-CoV-2 Assay in all of the samples analyzed, as demonstrated by lower Ct values (Figure 4B). Therefore, this annealing temperature was selected in the following experiments as the optimal annealing temperature for JRC-CoV-UCE.

We then tested whether the JRC-CoV-UCE assay would also perform well as a one-step ddRT-PCR assay and whether it could possibly be employed to determine the viral load in a given sample. To this end, one Omicron-positive clinical sample was used (BA.1.17.2 sub-lineage). The JRC-CoV-UCE.1 and JRC-CoV-UCE.2 assays were run together as a duplex assay. As a control, we used the CDC-N assay [37]. Similar results were obtained with all three of the assays indicating that JRC-CoV-UCE can be used in both RT-PCR and ddRT-PCR formats (Figure 4C).

We then calculated the analytical limit of detection (LoD) and the linearity of the signal for the JRC-CoV-UCE assay in comparison with several WHO-recommended assays. Serial dilutions of a characterized clinical sample, positive for Omicron for which the viral load was previously measured by ddRT-PCR, were used as a template. The JRC-CoV-UCE.1 and JRC-CoV-UCE.2 assays were run as a duplex RT-PCR assay but analyzed singly. Concentrations up to 5 cp/µL were detected by the JRC-CoV-UCE assay, with -UCE.2 approximately ten times more sensitive than UCE.1. The JRC-CoV-UCE assay performed similarly to the HKU-N and CDC-N assays and slightly better than the 2019-nCoV N1, 2019-nCoV N2, and E Sarbeco assays. The WH-NIC assay was confirmed to be the less sensitive assay, with it failing to detect Omicron at concentrations below 400 viral genome copies per reaction, thus resulting in it being about 100 time less efficient than the other assays. The signal for JRC-CoV-UCE was linear between 5 to 50,000 cp/reaction. All together, these results indicate that the JRC-CoV-UCE assay can detect Omicron with a satisfactory level of sensitivity (Figure 4D). A similar LoD was calculated for the JRC-CoV-UCE assay when a synthetic Omicron RNA genome was used as a template for the RT-PCR reaction (Appendix A).

### 3.3. JRC-CoV-UCE Assay Performance in Clinical and Wastewater Samples

We then studied the versatility of JRC-CoV-UCE using characterized clinical swab samples containing Wuhan H-1, Alpha, Gamma, Delta, Omicron BA.1, BA.2, BA.3, BA.4, and BA.5 variants. As controls, clinical samples negative for SARS-CoV-2 were used (*n* = 10). In agreement with the results obtained with synthetic RNA genomes, JRC-CoV-UCE was able to specifically detect all SARS-CoV-2 variants very efficiently with a performance that was very comparable to that of the Seegene AllPlex assay (Figure 5A). No false positive results were obtained in SARS-CoV-2-negative samples.

Analysis repeated with a larger cohort of clinical samples (*n* = 40), containing Omicron at different viral loads or negative controls (samples negative for SARS-CoV-2, *n* = 20), confirmed that the JRC-CoV-UCE assay efficiently detects Omicron in all of the samples analyzed, including in those where the virus was present in low copy numbers (Figure 5B). Consistent with previous results, no false positive results were reported using SARS-CoV-2-negative samples, thus representing the high specificity of the method. Similar results were also obtained by analyzing the same samples in a second laboratory using a different RT-PCR instrument (for details on the instruments used see the Section 2).

Surveillance of SARS-CoV-2 and its variants in wastewater is less resource-intensive than large-scale clinical testing and can provide rapid and reliable sources of information of virus circulation in a given population. Researchers can test wastewater samples for specific viral genetic signatures (e.g., by RT-PCR) without the need to perform extensive sequencing. We tested the JRC-CoV-UCE method in comparison with the 2019-nCoV N2 and E Sarbeco methods on SARS-CoV-2 RNAs extracts from 18 different positive and three negative wastewater samples. These samples were collected between October and November 2021 when the Delta variant was predominant in the population and the Omicron variant was just reported for the first time in South Africa (https://www.who.int/news/item/26-11-2021-classification-of-omicron-(B.1.1.529)-sars-cov-2-variant-of-concern, accessed on 5 October 2022). Therefore, in parallel, we also analyzed these samples using the OmMet method, which is highly specific for the detection of Omicron [22]. JRC-CoV-UCE was able to detect with high specificity of all of the positive samples with a performance level slightly superior to that of the two WHO-recommended assays, 2019-nCoV N2 and E Sarbeco, that were used here as control reference methods (Figure 6C). Interestingly, two wastewater samples collected at Amsterdam Schiphol airport on 16 November and at Frankfurt am Main airport on 23 November tested positive for Omicron before the WHO officially designated Omicron as a VOC, thus providing evidence that Omicron had already arrived in Europe before the WHO’s announcement (Figure 6). These results also highlight the role of wastewater surveillance, in conjunction with highly specific and sensitive assays, for the early detection of new variants in a certain population.

## 4. Discussion

The Omicron variant in comparison to the Wuhan H1 prototype has acquired a large number of mutations throughout its genome. These mutations may potentially affect the performance of routinely used PCR and antigen-detection tests with the possibility of some viral genomes escaping detection. Our in silico analysis predicted that most of the WHO-recommended assays, despite having been developed prior to the emergence of Omicron, continue to detect Omicron with high efficiency, including the currently dominant BA.4 and BA.5 lineages. This reflects the high degree of conservation of the regions recognized by the primers/probe sets of these assays (Figure 3). However, we found that the WH-NIC-N assay developed by the NIH of Thailand [41] had a dramatic decrease in performance against Omicron in comparison with the Wuhan H1 prototype RNA. It is therefore likely that the assay would fail to detect Omicron in samples containing low viral concentrations and when viral RNA extraction is not optimal.

Mutations falling into the primers/probe sequences were also found for three other assays, namely 2019-nCoV N1, China-CDC N, and E Sarbeco [34,35,36,37]. We found that these assays still perform efficiently against Omicron. Nevertheless, it is important to continue monitoring their performance in the event that the virus would acquire new mutations falling within their target sequences.

The effects of mismatches on the PCR amplification kinetics and efficiency [45] are difficult to predict and are influenced by numerous factors, including the length of the oligonucleotides and the number, the nature, and position of the mismatches, as well as physical (the temperature of hybridization) and chemical (the presence of co-solvents and the concentrations of oligonucleotides and monovalent and divalent cations) parameters [46,47]. These factors do not impair the effectiveness of the PCR with the same magnitude. For instance, primer mismatches at the 3′ ends impair the PCR reaction in a stronger manner than a few mismatches located elsewhere in the primers. Looking at our results, one may suppose that the single mismatch found between the forward primer of the E Sarbeco method and its target Omicron consensus sequence (Figure 1) has a minor impact on the PCR’s performance. Indeed, our results confirmed that the method detects Wuhan H1 and the Omicron genome with similar efficacy (Figure 2). Nevertheless, without experimental evidence, we cannot completely exclude a negative effect on PCR performance from the other mismatches found for other assays against some of the SARS-CoV-2 sub-lineages (Figure 1). The performance of diagnostic PCR assays can therefore be impacted by SARS-CoV-2 variability as this is dependent on the complementarity between PCR primers/probes and viral target templates [48]. Changes in performance may occur especially for those assays with a single genetic target due to the presence of a mutation in the target area. Therefore, assays with multiple targets in different regions of the viral genome have a reduced risk to fail in detecting new emerging SARS-CoV-2 variants and are recommendable [49].

There are few doubts that the virus will continue to evolve and new variants may emerge [16]. Mutations occur randomly and are selected if they are beneficial for the viral fitness e.g., increased transmissibility and/or the ability to evade the host immunity. It is possible that SARS-CoV-2 will acquire further genomic changes falling into the sequences recognized by the primers and probes of these assays and therefore reducing their diagnostic accuracy. An example is the variant of interest BA.2.75, which shows 75 base changes and four deletions of nine bps or more. Fortunately, these mutations do not fall within the target sequences of the primers/probes of WHO-recommended RT-PCR and JRC-CoV-UCE assays. Similarly, analysis with the most recent BQ.1 and XBB did not reveal any further mismatches between the oligonucleotides of the assays and their consensus sequences which may predict a drop in their performance. Therefore, these assays are expected to detect with similar efficiency these more recent sub-linages. Nevertheless, it is important on the one hand to continue our efforts in sequencing the viral genome to closely monitor SARS-CoV-2’s evolution, and on the other hand, to timely check the performance of routinely used assays. The in silico PCR simulation approach used in this study, may be one of the tools that could be employed to predict whether a certain mutation may potentially impact test performance. However, as also shown by our results, any predicted drop in diagnostic performance should then be confirmed in the laboratory with clinical samples.

A recent experimental evolution study estimated that the SARS-CoV-2 genome mutates at the rate of 1.25 × 10^−6^/nt/replication cycle with the S gene mutating at least five-fold higher than the genomic average [50]. However, not all of the regions of the viral genome mutate at the same rate. There are regions that are more conserved than others, most likely because their integrity is crucial for the virus’s life cycle. Although most of the WHOrecommended tests are still reliable against Omicron, to minimize the impact of viral mutations for the monitoring of SARS-CoV-2, we decided to increase the portfolio of SARS-CoV-2 assays with two new assays targeting the most highly conserved regions in the SARS-CoV-2 genome. We found two regions, located within the *Orf1a* gene in the part encoding NSP3 and NSP9 proteins, fit for our purpose. To the best of our knowledge, this is the first time that the NSP3 region has been selected in SARS-CoV-2 PCR diagnostics, while the NSP9 region was already a target for the WHO-recommended assay RdRp-nCoV-IP2 developed at the Institute Pasteur, France [40].

The JRC-CoV-UCE assay was optimized to run as a duplex, and laboratory validation carried out with synthetic RNA genomes and historical clinical and wastewater samples shows high efficiency in the detection of all of the VOCs characterized so far.

Due to the high degree of conservation of the target sequences, we anticipate that the assay will continue to perform with high accuracy with the next generation of SARS-CoV-2 variants. However, its LoD should be verified in case of the emergence of new variants. To run the assay as a duplex or as a multiplex in combination with the other WHO-recommended assays, targeting other conserved regions within the SARS-CoV-2 genome, will further minimize the risk that new variants escape detection.

The JRC-CoV-UCE assay can be used to monitor the emergence of a potential novel VOC by coupling their usage with those of VOC-specific detection methods. For example, as Omicron is presently the dominant circulating VOC, a collected sample can be quickly screened by JRC-CoV-UCE and tested simultaneously with an Omicron-specific detection method, such as OmMet [25]. If their relative ratios are in the same range, it is likely that the sample contains predominantly Omicron-related variants. If, on the contrary, JRC-CoV-UCE detects the presence of SARS-CoV-2 while the VOC-specific method signal is found to be lower or absent, then there is a high probability that a “not-Omicron-related” variant is present in the analyzed sample. This specific sample should then be further characterized, e.g., viral RNA sequencing to identify possible new mutations within the viral genome.

Wastewater surveillance provides a cost-effective, rapid, and reliable source of information on the presence, prevalence, and spread of SARS-CoV-2 and other pathogens in a community. Wastewater surveillance has been demonstrated to be valuable in combating infectious diseases such as Hepatitis A, Hepatitis E, paralytic polio, and more recently COVID-19 [51,52]. Growing evidence indicates that the prevalence and circulation profiles of SARS-CoV-2 variants in wastewater closely correlate with clinical data [53].

Wastewater surveillance detects the presence of SARS-CoV-2 shed by people with and without symptoms and is therefore not directly dependent on people’s access to healthcare, seeking healthcare when they become ill, or on the availability of COVID-19 tests. A number of different laboratories have shown that detection of the virus in wastewater samples preceded its detection at the clinical level [44]. We confirmed the presence of Omicron in wastewater samples from Amsterdam Schiphol airport and Frankfurt am Main airport before Omicron was declared as a VOC by the WHO (Figure 6). These results reinforce the idea of using wastewater surveillance as a preventive or early warning system for the long-term monitoring of SARS-CoV-2 as well as early identification of viral circulation in the population.

Another important outcome of this study was to find out that JRC-CoV-UCE, although optimized as a RT-PCR assay, can be run as well as in the ddRT-PCR format without any further optimization. It is known that ddPCR enables an absolute quantification of nucleic acid molecules providing results directly as copy numbers per µL instead of quantification cycles (Cq) (also termed cycle threshold (Ct) values) of the qPCR assays. Cq (or Ct) values are dependent on the instrument and software used [54]. These data require the use of traceable calibrators, such as certified reference materials, to make the standard curves needed to convert them into absolute numbers and to assure the intra-laboratory comparability of the results. The ddPCR has been widely used to determine the copy numbers of DNA and RNA viruses including SARS-CoV-2 [54,55,56,57,58,59,60]. It offers greater robustness against PCR inhibitors found in more difficult samples with a variable and complex matrix such as wastewater samples [56,57,58,59] and is slightly more sensitive than qPCR [54,60]. The fact that the JRC-CoV-UCE assay can be converted into a ddRT-PCR assay is particularly relevant in view of its use in wastewater surveillance where it is crucial to precisely monitor virus loads in a certain community as a function of time. Unfortunately, the lack of ddPCR equipment and expertise, not always available in standard diagnostic laboratories, and the relatively higher costs limit the use of ddPCR-based protocols in routine applications.

Tracking SARS-CoV-2 variants in communities is therefore becoming pivotal to assure a rapid public health response. This has also been recently recognized by the leaders of the EU, the G7, the G20, and by the European Commission [61]. The European Commission recommends that Member States establish systematic surveillance of SARS-CoV-2 and its variants in wastewater and to include wastewater monitoring of the SARS-CoV-2 virus in national testing strategies as a complementary approach to existing COVID-19 surveillance systems and clinical testing [62]. A common approach to screen and select wastewater samples for further characterization would be highly beneficial. The JRC-CoV-UCE assay coupled with other VOC-specific RT-PCR methods may therefore contribute to this aim and support the collaborating parties and entities of the EU Sewage Sentinel System for SARS-CoV-2 [63].

In conclusion, as stated by the ECDC recently, it is crucial that laboratories continue to monitor SARS-CoV-2’s evolution and ensure that laboratory testing systems are performing adequately for the circulating viruses [63]. We have provided a new assay for the detection of SARS-CoV-2 that may contribute to the containment of the virus and in managing its spread.

## Figures and Tables

**Figure 1 viruses-15-00206-f001:**
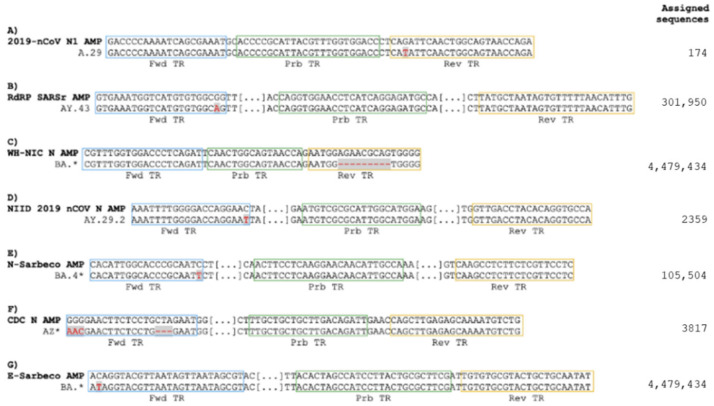
Examples of mismatches found between oligonucleotides of the indicated WHO-recommended RT-PCR assays and their viral genome consensus sequences. Examples of lineage-specific variants (**A**–**G**) that can affect the detectability of the WHO-recommended methods are shown. For each example, the alignment between the primers/probes’ target regions on the reference amplicon (AMP) and the corresponding target regions on the lineage consensus sequence is illustrated. The forward (Fwd), probe (Prb), and reverse (Rev) target regions (TR) are boxed in blue, green, and yellow, respectively. Mismatches are highlighted in red on a gray background. For each lineage (or group of lineages when the name is marked with “*”), the number of GISAID-assigned sequences is reported on the right.

**Figure 2 viruses-15-00206-f002:**
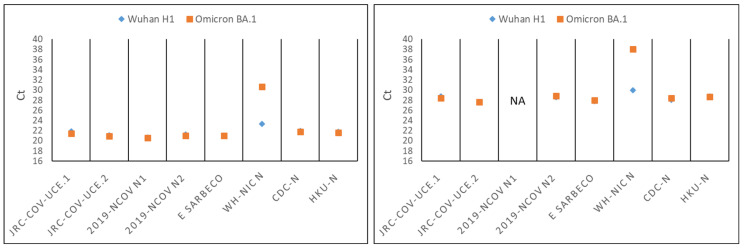
The WH-NIC N assay has reduced abilities to identify Omicron. The performances of the indicated WHO-recommended assays and the JRC-CoV-UCE. 1 and JRC-CoV-UCE.2 assays (see next paragraphs) in detecting the Wuhan H1 prototype or Omicron variant were compared. Either 5 × 10^5^ or 5 × 10^3^ copies of Wuhan H1 or Omicron synthetic viral genomic RNA per RT-PCR reaction were used in the left or right panel, respectively. Dots represent the average Ct values with the relative standard deviation of a typical experiment performed in triplicate. NA: not available.

**Figure 3 viruses-15-00206-f003:**
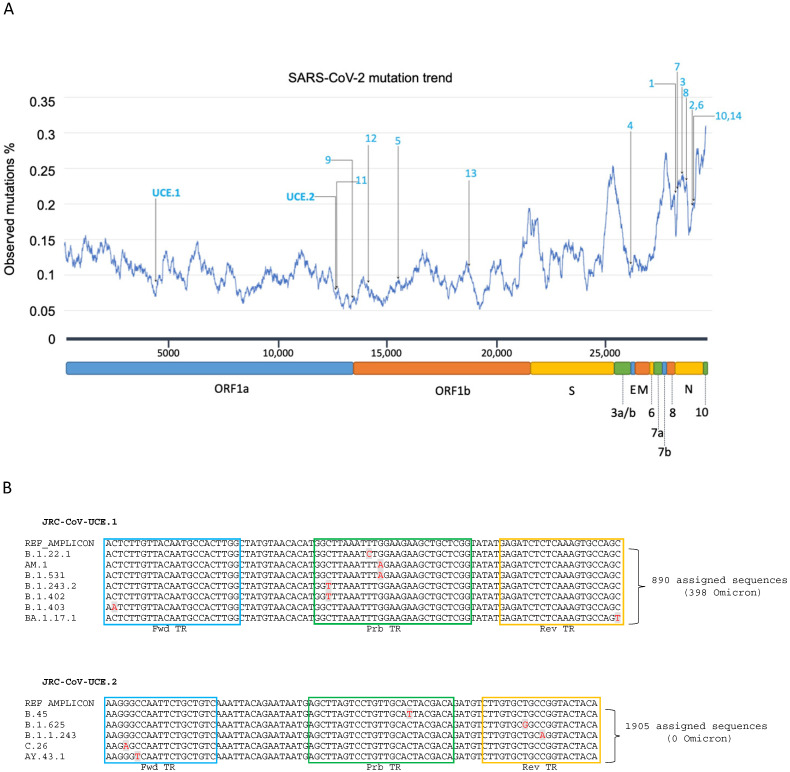
Design of JRC-Cov-UCE assay. (**A**) Selection of conserved regions within the SARS-CoV-2 genome. The number of mutations present in the regions targeted by the different primers and probes of the indicated RT-PCR assays were calculated using all of the SARS-CoV-2 genome consensus sequences. 1: 2019-nCoV N1; 2: 2019-nCoV N2; 3: 2019-nCoV N3 [34,35]; 4: E Sarbeco [36]; 5: RdRP SARSr [36]; 6: N Sarbeco [36]; 7: WH-NIC N [41]; 8: CDC-N [37]; 9: CDC-ORF1ab [37]; 10: NIID 2019 nCOV N [39]; 11: nCoV IP2 [40]; 12: RdRp gene/nCoV IP4 [40]; 13: HKU-ORF1b-nsp14 [38]; 14: HKU-N [38]. Abbreviations: UCE.1:and UCE.2 ultra conserved elements targeted by the JRC-CoV-UCE assay; ORF: open reading frame; S: Spike; E: envelope; M: membrane; N: nucleocapsid; NSP: non structural protein. (**B**) Sequence alignment. The alignments between the primers/probes of JRC-CoV-1 and JRC-CoV.UCE.2 were performed against all 1815 SARS-CoV-2 consensus sequences. Only the consensus sequences in which mismatches were found (in red) are represented. This corresponds to a small minority of the total SARS-CoV-2 genomic sequences analyzed. The forward (Fwd), probe (Prb), and reverse (Rev) target regions (TR) are boxed in blue, green, and yellow, respectively.

**Figure 4 viruses-15-00206-f004:**
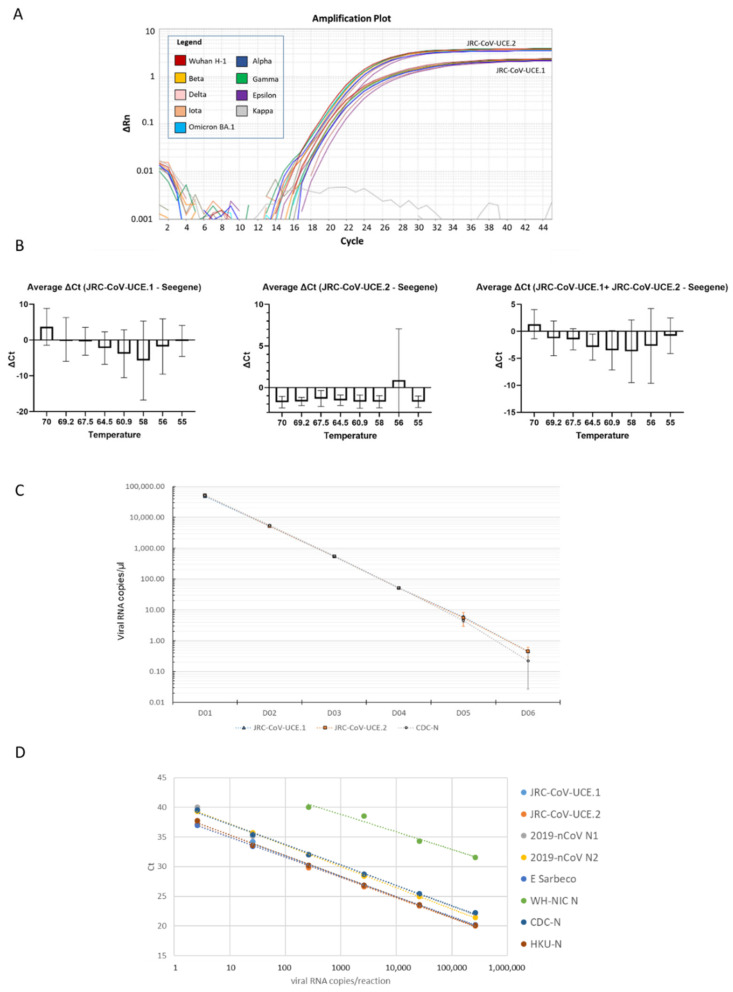
JRC-CoV-UCE performance assessment and optimization. (**A**) The duplex JRC-CoV-UCE indistinctly detects all of the tested SARS-CoV-2 variants which have been circulating from the beginning of the pandemic. Real-time RT-PCR were performed using synthetic RNA genomes (5 × 10^5^ copies/reaction) of the Wuhan H-1 prototype or indicated variants. Both JRC-CoV-UCE.1 and JRC-CoV-UCE.1 detected all of the variants with similar efficiencies.(**B**) Determination of optimal annealing temperature. JRC-CoV-UCE was tested at eight different annealing temperatures (measured in degrees Celsius) and compared with the results obtained using the Seegene’s AllPlex SARS-CoV-2 assay. The plots represent the differences in the Seegene Allplex average Ct values from (left) JRC-CoV-UCE.1, (middle) JRC-CoV-UCE.2, and (right) the average of both JRC-CoV-UCE.1 and JRC-CoV-UCE.2 Ct values, with standard deviation bars. (**C**) ddRT-PCR. The JRC-CoV-UCE and CDC-CN [37] assays were run as ddRT-PCR assays using a clinical sample containing Omicron (BA.1) as a template. Viral load was quantified and expressed as viral RNA copies/µL). D01–D06 on the X-axis indicate the 1:10 serial dilutions carried out for the analysis. (**D**) LoD calculation. After quantification of the viral load by ddRT-PCR, the clinical sample was serial diluted and then analyzed by RT-PCR with the indicated assays.

**Figure 5 viruses-15-00206-f005:**
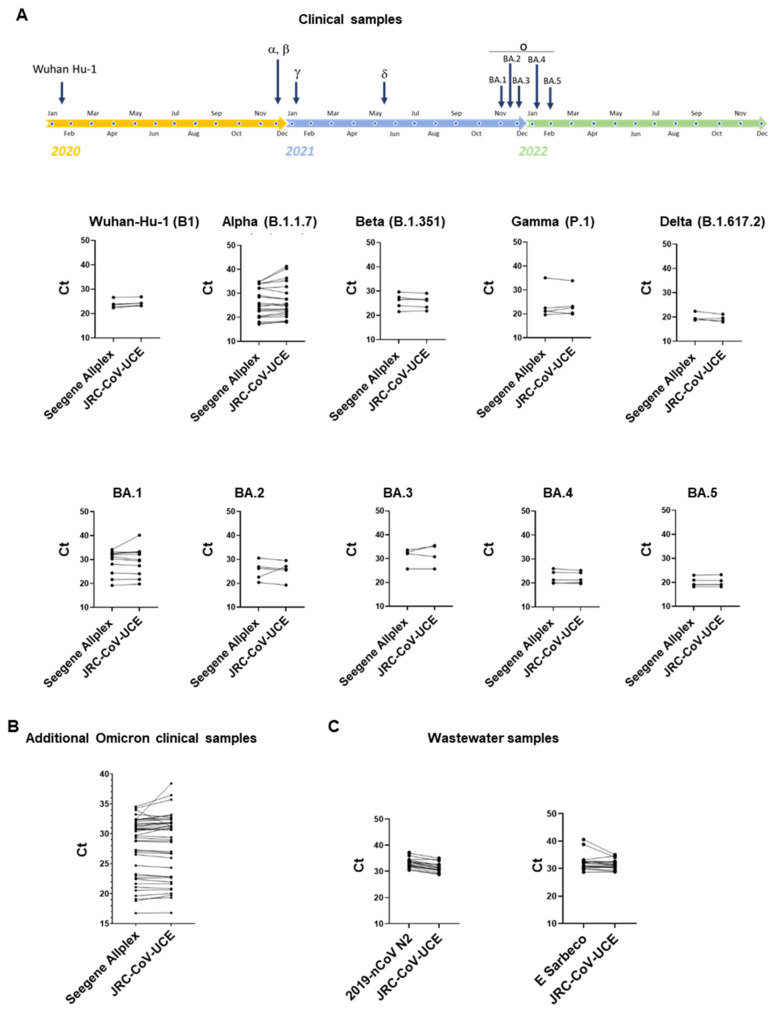
The JRC-CoV-UCE duplex assay is able to detect all of the SARS-CoV-2 variants in clinical and environmental samples. (**A**) Timeline showing the emergence of SARS-CoV-2 VOCs, as stated by the WHO (top). The clinical samples were tested with both JRC-CoV-UCE and the Seegene’s AllPlex assay, and the average Ct values are shown for the matched samples. (**B**) The JRC-CoV-UCE duplex assay efficiently detects Omicron of varying viral loads. Omicron-positive clinical samples were tested with both JRC-CoV-UCE and the Seegene’s AllPlex assay. (**C**) The wastewater samples were analyzed with the 2019-nCoV N2 (N2, left) and E Sarbeco (right) assays and compared with the JRC-CoV-UCE assay.

**Figure 6 viruses-15-00206-f006:**
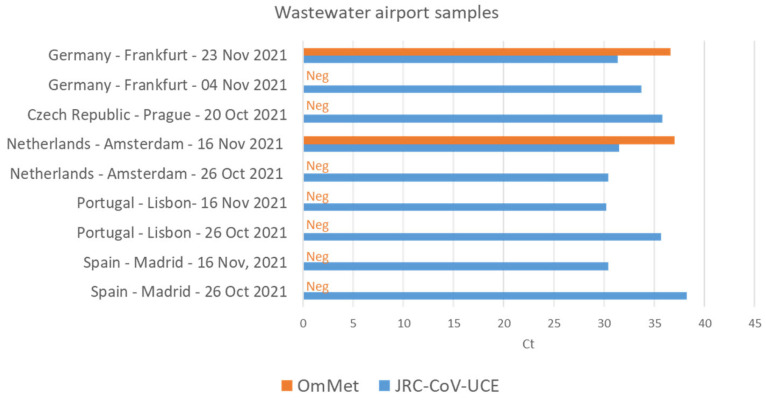
Early detection of Omicron in airport wastewater samples. Samples from the indicated airports were analyzed by RT-PCR with both the JRC-CoV-UCE and the Omicron-specific OmMet [22] assays. SARS-CoV-2 (most likely the Delta variant, which was highly predominant at that time) was identified in all of the samples while Omicron was only detected at lower concentrations (higher Ct numbers) in two samples from airports in Frankfurt and Amsterdam from the middle and end of November 2021. The presence of Omicron was confirmed by genome sequencing [44]. Neg: no detection of Omicron by using the OmMet assay.

**Table 1 viruses-15-00206-t001:** Primers and TaqMan^®^ probe sequences with relative concentrations used, reference amplicon generated, and corresponding genomic location on the SARS-CoV-2 genome.

Assay	Reference Primers and Probes	Reference Amplicon (Size, bp)	SARS-CoV-2 Target [Genomic Location] ^1^
JRC-CoV-UCE.1	For: 5′-ACTCTTGTTACAATGCCACTTGG-3′ (900) Rev: 5′-GCTGGCACTTTGAGAGATCTC-3′ (900) P: 5′-VIC-GGCTTAAATTTGGAAGAAGCTGCTCGG-QSY-3′	ACTCTTGTTACAATGCCACTTGGCTATGTAACACATGGCTTAAATTTGGAAGAGCTGCTCGGTATATGAGATCTCTCAAAGTGCCAGC (89)	Orf1a(NSP3) [4595–4683]
JRC-CoV-UCE.2	For: 5′-AAGGGCCAATTCTGCTGTC-3′ (900) Rev: 5′-TGTAGTACCGGCAGCACAAG-3′ (900) P: 5′-ABY-AGCTTAGTCCTGTTGCACTACGACA-QSY-3′	AAGGGCCAATTCTGCTGTCAAATTACAGAATAATGAGCTTAGTCCTGTTGCACTACGACAGATGTCTTGTGCTGCCGGTACTACA (85)	Orf1a(NSP9) [12658–12742]

^1^ According to the Wuhan reference sequence NC_045512.2.

**Table 2 viruses-15-00206-t002:** List of WHO-recommended methods with respective primers and conditions. FAM: 6-carboxyfluorescein; BHQ1: Black Hole Quencher 1 (Biosearch Technologies, Inc., Novato, CA, USA); BBQ-650: BlackBerry Quencher 650 (Berry & Associates, Dexter, MI, USA); TAMRA: Tetramethylrhodamine (Applied Biosystems, Waltham, MA, USA).

Developer	NAAT Name	Reference Primers and Probes (Concentration, nM)	Annealing Temperature (°C)
US CDC, USA	2019-nCoV N1	For: 5′-GACCCCAAAATCAGCGAAAT-3′ (500) Rev: 5′-TCTGGTTACTGCCAGTTGAATCTG-3′ (500) P: 5′-FAM-ACCCCGCATTACGTTTGGTGGACC-BHQ1-3′ (125)	55
US CDC, USA	2019-nCoV-N2	For: 5′-TTACAAACATTGGCCGCAAA-3′ (500) Rev: 5′-GCGCGACATTCCGAAGAA-3′ (500) P: 5′-FAM-ACAATTTGCCCCCAGCGCTTCAG-BHQ1-3′ (125)	55
Charité, DE	E Sarbeco	For: 5′-ACAGGTACGTTAATAGTTAATAGCGT-3′ (400) Rev: 5′-ATATTGCAGCAGTACGCACACA-3′ (400) P: 5′-FAM-ACACTAGCCATCCTTACTGCGCTTCG-BBQ650-3′(200)	58
NIH, TH	WH-NIC N	For: 5′-CGTTTGGTGGACCCTCAGAT-3′ (800) Rev: 5′-CCCCACTGCGTTCTCCATT-3′ (800) P: 5′-FAM-CAACTGGCAGTAACCA- BHQ1-3′ (200)	55
CDC, CN	CDC-N	For: 5′-GGGGAACTTCTCCTGCTAGAAT-3′ (400) Rev: 5′-CAGACATTTTGCTCTCAAGCTG-3′ (400) P: 5′-FAM-TTGCTGCTGCTTGACAGATT-TAMRA-3′ (200)	60
HKU, HK	HKU-N	For: 5′-TAATCAGACAAGGAACTGATTA-3′ (500) Rev: 5′-CGAAGGTGTGACTTCCATG-3′ (500) P: 5′-FAM-GCAAATTGTGCAATTTGCGG-BBQ1-3′ (250)	60

## Data Availability

All sequencing data from Luxembourg are publicly available on GISAID and provided in the Appendix A. All other raw data are available upon request.

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
