# Peer review of "New RT-PCR Assay for the Detection of Current and Future SARS-CoV-2 Variants"

_viruses, 2023, doi:10.3390/v15010206_

Round 1

Reviewer 1 Report

It is a well written paper evaluating the performance of the 14 WHO recommended real-time RT-PCR assays currently in use for the detection of SARS-CoV-2 and also developing a new duplex real-time RT-PCR assay

Author Response

"It is a well written paper evaluating the performance of the 14 WHO recommended real-time RT-PCR assays currently in use for the detection of SARS-CoV-2 and also developing a new duplex real-time RT-PCR assay"

Authors: Thank you very much for your positive feedback. Well appreciated.

Reviewer 2 Report

At presented manuscript, authors developed a new duplex real-time RT-PCR assay based on the amplification of two ultra-conserved elements present within the SARS-CoV-2 genome. Main speculation is that the new duplex assay successfully detects all tested SARS-CoV-2 variants of concern (especially Omicron sublineages).

The presented work certainly contains new and useful information. Nevertheless, having a clear emphasis on the methodological aspects of coronavirus detection, it is the methodological and analytical points of this work that raise a number of questions.

Information about the in silico procedure for evaluating primers is clearly insufficient. The authors refer here to their previous work, but I do not find these details there either (except for the ancient 2008 reference to THERMOBLAST). This question is serious, since in silico evaluation of primers can vary and often requires connection with experiment and experimental data. Actually, the authors themselves demonstrate this in their work, although they do not discuss it properly. - «The in silico prediction was confirmed experimentally. The WH-NIC N assay showed a reduced ability to detect Omicron RNA in comparison with the Wuhan H1 prototype (approximately 8 Ct of difference using the same input amounts of synthetic RNA viral genomes) (Figure 3). On the contrary, the China-CDC-N and E Sarbeco assay, despite presenting mismatches in their primers against the corresponding Omicron consensus sequence, recognized both Omicron and Wuhan H1 sequences with similar efficiency».

It is well known that the effect of the presence of mismatches in the primer on the amplification efficiency largely depends on their type, position relative to the 3'-end of the primer, the length of the primer, the ratio of Tm and Ta. Without considering these parameters, the meaning of the estimates eludes.

Do I understand correctly that the authors consider the total number of mismatches in the assortment of sites under the primer of all the sequences they analyzed? If yes, then the figure is not very informative and may even be misleading if the mismatches are localized at the 5' end of the primer. Figure 2 is even more uninformative. It would be better if the authors showed the alignment of all the analyzed primers/sample systems and gave a quantitative consensus. In our practice, we observed no decrease in the effectiveness of PCR in the presence of even two mismatches in the primer (at positions -8 and -10, the total length of the primer is 24 Tm =69oC). Such cases are quite common.

The analytical characterization section (LOD and LOQ) is written sparingly, there is not enough technical, methodological and statistical information.

The logic of comparison with a commercial test system (AllPlex SARSCoV-2 Assay, Seegene) is not very clear, authors do not have their primer sequences, the properties of the mixture for reverse transcription and amplification apparently differ for used Taqman Fast Virus 1-Step Master Mix. Wouldn't it be more correct to take some primers/probes from the ones analyzed by them in silico for the purity of the experiment?

At the same time, if you look at Figure 6, it shows that a significant part of the samples used for validation have low Ct and high viral load. It is clear that different test systems have the main difficulties when working with samples over 30 Ct. How many of these have been analyzed and compared with Seegene? Is this enough for the conclusions made in the work?

Author Response

At presented manuscript, authors developed a new duplex real-time RT-PCR assay based on the amplification of two ultra-conserved elements present within the SARS-CoV-2 genome. Main speculation is that the new duplex assay successfully detects all tested SARS-CoV-2 variants of concern (especially Omicron sublineages).

The presented work certainly contains new and useful information. Nevertheless, having a clear emphasis on the methodological aspects of coronavirus detection, it is the methodological and analytical points of this work that raise a number of questions.

Authors: Thank you very much for your positive feedback. We were glad to know that you found our study novel and useful. We hope that we have addressed all your questions in the revised version of the manuscript.

Information about the in silico procedure for evaluating primers is clearly insufficient. The authors refer here to their previous work, but I do not find these details there either (except for the ancient 2008 reference to THERMOBLAST). This question is serious, since in silico evaluation of primers can vary and often requires connection with experiment and experimental data. Actually, the authors themselves demonstrate this in their work, although they do not discuss it properly. - «The in silico prediction was confirmed experimentally. The WH-NIC N assay showed a reduced ability to detect Omicron RNA in comparison with the Wuhan H1 prototype (approximately 8 Ct of difference using the same input amounts of synthetic RNA viral genomes) (Figure 3). On the contrary, the China-CDC-N and E Sarbeco assay, despite presenting mismatches in their primers against the corresponding Omicron consensus sequence, recognized both Omicron and Wuhan H1 sequences with similar efficiency».

Authors: Thank you for your constructive criticism. In the new version of the manuscript, we have largely revised the Materials and Methods section providing new details on how we designed the primers of our JRC-CoV-UCE RT-PCR assay and performed the in silico PCR simulation (new paragraphs 2.1-2.2 and 2.3). We also agree with your comment that “in silico evaluation of primers requires connection with experiment and not always is predictive of performance (please see revised Discussion section, last paragraph page 16 and second paragraph page 17).

It is well known that the effect of the presence of mismatches in the primer on the amplification efficiency largely depends on their type, position relative to the 3'-end of the primer, the length of the primer, the ratio of Tm and Ta. Without considering these parameters, the meaning of the estimates eludes.

Authors: We agree with your statement and took into consideration your comment in the discussion section (last paragraph page 16).

Do I understand correctly that the authors consider the total number of mismatches in the assortment of sites under the primer of all the sequences they analyzed? If yes, then the figure is not very informative and may even be misleading if the mismatches are localized at the 5' end of the primer. Figure 2 is even more uninformative. It would be better if the authors showed the alignment of all the analyzed primers/sample systems and gave a quantitative consensus. In our practice, we observed no decrease in the effectiveness of PCR in the presence of even two mismatches in the primer (at positions -8 and -10, the total length of the primer is 24 Tm =69oC). Such cases are quite common.

Authors: We followed your suggestion and showed the alignments for all analysed primers/probe and the SARS-CoV-2 consensus sequences (new Figure 1 and Supplementary data file). We moved the previous Fig. 1 and 2 to Supplementary materials (Figure S1 and S3). 

The analytical characterization section (LOD and LOQ) is written sparingly, there is not enough technical, methodological and statistical information.

Authors: We created a new paragraph in the Materials and Methods section explaining how LoDs were calculated. We also performed a new set of experiments to calculate the LoD of our JRC-CoV-UCE method in comparison with the LoDs of several WHO recommended assays using a clinical sample for the PCR reactions. To this end, we first measured the viral loads in the clinical sample we used by droplet digital RT-PCR. By doing so, we discovered that our JRC-CoV-UCE assay also performed well as a one step ddRT-PCR assay without the need of further optimization. We are happy to show these results in the new revised Figure 4 (panel C and D). We moved the previous LoD calculation performed with synthetic viral genome to supplementary materials (Figure S4).

The logic of comparison with a commercial test system (AllPlex SARSCoV-2 Assay, Seegene) is not very clear, authors do not have their primer sequences, the properties of the mixture for reverse transcription and amplification apparently differ for used Taqman Fast Virus 1-Step Master Mix. Wouldn't it be more correct to take some primers/probes from the ones analyzed by them in silico for the purity of the experiment?

Authors: The AllPlex SARS-CoV-2 Assay from Seegene was always run in tandem as a control with the JRC-CoV-UCe assay for clinical samples at LNS as this is the standard kit used in diagnostics for Luxembourgish national reporting. The method is considered one of “ first in class” methods targeting four different viral sequences, and therefore we believe it offers an adequate control for testing the new RT-PCR assay. We are happy to note that our assay performs similarly to this one. Nevertheless, to address your point, in the revised version of the manuscript, we compared the sensitivity (LoDs) of JRC-CoV-UCE method with that of several WHO recommended assay (Figure 4 D)showing that the sensitivity of our method is similar (or even slight better) than that of the assays tested.

At the same time, if you look at Figure 6, it shows that a significant part of the samples used for validation have low Ct and high viral load. It is clear that different test systems have the main difficulties when working with samples over 30 Ct. How many of these have been analyzed and compared with Seegene? Is this enough for the conclusions made in the work?

Authors: In Figure 5 A and B, all samples were tested with both Seegene AllPlex and the JRC-CoV-UCE methods. Indeed, as many assays can fail when it comes to samples with high Ct values due to sensitivity, we found it important to also test samples with low viral loads (high Ct value)  (Fig 5 B). We analysed at least 10 clinical samples with low viral loads (25% of the total) The results did not have any false negatives (all samples were detectable with both methods). Furthermore, the wastewater samples we analysed all had low viral concentrations (more than 30 Ct) (Fig. 5 C). In agreement with previous results, our assay performed similarly than E Sarbeco and 2019 nCoV-N2 assays used here as control reference methods. Altogether, these results support the conclusions of the manuscript, including that JRC-CoV-UCE is a universal, highly sensitive method. We think that the number of samples with low viral concentrations we used for the experiments is sufficient to draw these conclusions.

Reviewer 3 Report

The authors describe the performance evaluation of the 14 WHO recommended real-time RT-PCR assays currently in use for the detection of SARS-CoV-2, followed by the development of a new duplex real-time RT-PCR assay based on the amplification of two ultra-conserved elements present within the SARS-CoV-2 genome. They showed the applicability in clinical and wastewater samples. The authors suggest that this universal assay, aiming to detect all present and future SARS-Cov2 variants, combined with a previously designed assay for the Omicron variant, may provide timely indication of the emergence of a novel SARS-CoV-2 variant in a certain community and thereby aid public health interventions.

The universal assay would be of great value to the SARS-Cov-2 monitoring and diagnosis. The manuscript is clearly written (with a few exception, see comments below), with a proper and logical design of the experimental workflow.

Major comments:

1. Materials and methods is not totally clear and complete and should be extended:

- how was the in silico analysis performed to define the ultra-conserved regions? How were the in silico alignments done? Which sequences were taken for this? How many? How representative is this?

-how was the in silico PCR performed?

-the tested WHO assays should be added to supplementary information, together with their primer and probe sequences and their LOD. The latter is needed to evaluate the results on the comparisons made, where 2 concentrations have been used (5 x 105 or 5 x 103 copies of Wuhan H1 or Omicron synthetic viral genomic RNA per RT-PCR reaction).

- what is ‘very low number of mutations’? (selection criterion for the 500 bp regions). In figure 4, it is clear that the one assay that didn’t detect the omicron in ‘real’ is not the one with the highest mutations. So is this the best criterion to be used?

- ‘at least 10 assigned SARS-CoV-2 sequenced genomes’ - which genomes have been used? These should be defined.

- ‘Samples were sequenced using Illumina technologies’ – which samples? How was this done? Where is the data?

-why are so many different qPCR instruments used for different tests? How does this influence the results (as it seems that a same assay with a same sample was not run on different instruments (robustness) but more assay-instrument specific)?

2. Why is another annealing temperature used for the synthetic versus the ‘real’ samples? What is the impact? The optimization was done for omicron, but what will happen for other variants (as for the synthetic genomes, there was already another temperature used)? This should be clarified.

3. Fig. 1: where are these mutations located? As shown by the authors, the in silico predictions do not totally correspond to the ‘real’ results (even with mutations, the assays work) – this is linked to the position of the mutation, and this information is not available and should be added. The method used by the authors for in silico analysis is based on number of mismatches only, without taking other factors into account (as was done in other studies with more complex tools used for in silico performance analysis). This limitation should be acknowledged in the discussion.

4. The LOD of the UCE assay was tested on the omicron synthetic genome, using the ‘optimized’ annealing temperature. How will this LOD ‘hold’ for the other current and future variants? This should be discussed.

5. ‘Interestingly, two wastewater samples collected at Amsterdam Schiphol and Frankfurt Main airports tested positive for Omicron two weeks prior to the WHO officially designed Omicron a VOC, providing evidence that Omicron had already arrived in Europe before the WHO announcement (data not shown).’ Such important statement cannot be made without showing the data. So either these should be added, or the statement should be removed from results and discussion.

6. “We tested the JRC-CoV-UCE method in comparison with the 2019-nCoV N2 and E Sarbeco methods on SARS-CoV-2 RNAs extracts …”. Why were these 2 assays selected for comparison? This just be clarified. This might also give some more information (which is needed) as to which extend the UCE assays are the ‘first’ universal assay or whether other assays could also be a universal one. This should be added to the discussion. Similar to this ‘….while the NSP9 region was already a target for the WHO recommended assay RdRp-nCoV-IP2 developed at the Institute Pasteur, France’. Why wasn’t then this assay chosen for comparison? Wouldn’t this also be a universal assay?

7. The discussion should reflect on the use of digital PCR compared to RT-qPCR for wastewater applications.

8. The discussion is at some occasions a repetitions of the results. This should be carefully checked and modified (some suggestions have been made as to how the discussion should/could be extended and go beyond repeating the results).

 Minor comments:

·         Abstract: have these variants really been isolated? Seems rather difficult for a virus. They have been identified/sequenced. This sentence should be modified.

·         Abstract: to indicate that environmental is waste water.

·         Introduction: to add the state of the art of the performance evaluation of the WHO recommended assays, both in silico and in vitro. Other studies have already done similar investigations. These should be mentioned. In the discussion a part should be added how these relate to the current findings.

·         Fig 2. the legend and materials & methods should explain the ‘consensus seq’ versus the 'total seq’ results.

·         On which samples was the in silico PCR verified? Synthetic genomes?

·         Fig 3 right panel: why are there no dots for 2019-NCOV-N1?

·         Fig. 4: what is the SARS-cov-2 genome? The Wuhan genome? To what was the comparison done to identify the mutations?

·         Fig. 5: it is not clear form the figure if each variant was detected with both UCE assays (as it was a duplex).

·         Fig. 5C: why was the LOD performed for the single assays and not for the duplex?

·         ‘Based on these results, in accordance with WHO guidelines, we would also like to recommend to use diagnostic and wastewater tests that include multiple gene targets rather than single target assays to reduce at the minimum the risk that some SARS-CoV-2 would escape detection.’ The authors are not the first to recommended this. This should be made clear (reformulation the sentence + adding appropriate references).

·         ‘It is possible that SARS-CoV-2 will acquire further genomic changes falling into the sequences recognized by primers and probes of these assays and therefore reducing their diagnostic accuracy. An example is the variant of interest BA.2.75, which shows 75 base changes and four deletions of nine bps or more.’ Was this tested (in silico or in real) by the authors? Can this be added?

·         ‘The in silico PCR simulation approach used in this study, accurately predicted the performance’. This is not totally correct, as the ‘real’ results were different from the in silico results for some assays (better results obtained with ‘real’ assay, compared to in silico. See also comment above on this. The sentence should be reformulated.

·         ‘SARS-CoV-2 has acquired a large number of mutations throughout its genome.’ The Wuhan strain?

Author Response

The authors describe the performance evaluation of the 14 WHO recommended real-time RT-PCR assays currently in use for the detection of SARS-CoV-2, followed by the development of a new duplex real-time RT-PCR assay based on the amplification of two ultra-conserved elements present within the SARS-CoV-2 genome. They showed the applicability in clinical and wastewater samples. The authors suggest that this universal assay, aiming to detect all present and future SARS-Cov2 variants, combined with a previously designed assay for the Omicron variant, may provide timely indication of the emergence of a novel SARS-CoV-2 variant in a certain community and thereby aid public health interventions.

The universal assay would be of great value to the SARS-Cov-2 monitoring and diagnosis. The manuscript is clearly written (with a few exception, see comments below), with a proper and logical design of the experimental workflow.

Authors: We were very pleased to know that the Reviewer has found our manuscript relevant and clearly written. We hope that in the new version of the manuscript we were able to address all his/her reaming concerns.

Major comments:

  1. Materials and methods is not totally clear and complete and should be extended:

- how was the in silico analysis performed to define the ultra-conserved regions? How were the in silico alignments done? Which sequences were taken for this? How many? How representative is this?

-how was the in silico PCR performed?

Authors: In the new version of the manuscript, we have extensively revised the Materials and Methods and written three new paragraphs (2.1-2.2 and 2.3). We hope that the new paragraphs satisfactory answer the reviewer’s questions.

-the tested WHO assays should be added to supplementary information, together with their primer and probe sequences and their LOD. The latter is needed to evaluate the results on the comparisons made, where 2 concentrations have been used (5 x 105 or 5 x 103 copies of Wuhan H1 or Omicron synthetic viral genomic RNA per RT-PCR reaction).

Authors: We provide the information requested about primers and probe sequences in Table S1. Furthermore, we performed a new set of experiments in which we calculated the LoD of JRC-CoV-UCE in comparison with the LoDs of several WHO recommended assays (Fig. 4 using a clinical sample positive for Omicron (Figure 4 D). To this end, we first measured the viral loads in the clinical sample we used by droplet digital RT-PCR technology. By doing so, we discovered that our assay also performed well as a one step ddRT-PCR assay, without the need of further optimization. We are happy to show these results in the new revised Fig. 4 (panel C). We moved the previous LoD calculation to supplementary materials (Figure S4).

- what is ‘very low number of mutations’? (selection criterion for the 500 bp regions). In figure 4, it is clear that the one assay that didn’t detect the omicron in ‘real’ is not the one with the highest mutations. So is this the best criterion to be used?

Authors: We addressed the question raised by the reviewer in the new version of the Materials and Methods clarifying how we identified the ultra conserved elements within the SARS-CoV-2 genome (paragraph 2.2 Identification of SARS-CoV-2 ultra conserved elements and design of RT-PCR assays). In addition, we discussed that the number of mismatches cannot be the only criterion to predict qPCR performance. Numerous other factors, may have an impact on the performance of RT-PCR assays e.g. the length of the oligonucleotides, the number, the nature and position of the mismatches, the temperature of hybridisation, the presence of co-solvents, the concentrations of oligonucleotides, of monovalent and divalent cations etc...(last paragraph of the discussion at page 16). Furthermore, also addressing a criticism of Reviewer 1, we provide information about the position of the mismatches in the new Figure 1 and in Supplementary information.

- ‘at least 10 assigned SARS-CoV-2 sequenced genomes’ - which genomes have been used? These should be defined.

Authors: We clarified this issue in the new paragraph 2.1 of the Materials and Methods titled: “Generation of lineage consensus sequences”

- ‘Samples were sequenced using Illumina technologies’ – which samples? How was this done? Where is the data?

Authors: The samples that were used within this manuscript had been previously sequenced and the variant classification from the sequencing data is how we chose samples to be used for the tests within the manuscript. We have updated the text on this note to avoid confusion and we provide this information in Supplementary materials (Supplementary data file II).

-why are so many different qPCR instruments used for different tests? How does this influence the results (as it seems that a same assay with a same sample was not run on different instruments (robustness) but more assay-instrument specific)?

Authors: We used different qPCR instruments simply because the experiments were performed in collaboration at different laboratories. In particular, for the results shown in Fig. 5 A, two simultaneous runs were performed at LNS using two BioRad CFX96 Real-Time PCR Systems. One run was performed using the JRC-CoV-UCE protocol and the second with the AllPlex SARS-CoV-2 Assay, as a control. However, we addressed the inter-laboratory comparability of the results for samples presented in Figure 5B (40 Omicron samples). Exactly the same samples were analysed at both LNS and the JRC, sharing the same qPCR protocols but using two different instruments, the BioRad CFX96 Real-Time PCR System and a calibrated QuantStudio 7 Flex Real-Time PCR System. The results from both labs were comparable providing evidence of the robustness of the assay. This information has been added in the revised version of the manuscript in both the Materials and Methods and results sections.

  1. Why is another annealing temperature used for the synthetic versus the ‘real’ samples? What is the impact? The optimization was done for omicron, but what will happen for other variants (as for the synthetic genomes, there was already another temperature used)? This should be clarified.

Authors: We did not observe major difference running the assay with annealing temperature in the range 60-64.5 °C. This information has been added in the text at page 12 of the revised version of our manuscript. Only in the very first experiment (Fig. 4 A) using synthetic RNA genomes to verify our hypothesis that the assays would have recognized all SARS-CoV-2 variants with similar efficiency, the experiment was performed at an annealing temperature of 62 °C. The optimization of the annealing temperature was done to increase the performance of the JRC-CoV-UCE method, especially when it came to samples with high Ct values. This was done by comparing the performance of JRC-CoV-UCE and the Seagene.method (Fig. 4B). The optimization was done for Omicron because is the circulating variant at the date. We add a sentence in the discussion saying that even if we think that due to the high degree of conservation of the target sequences, the assay will continue to perform with high accuracy with the next generation of SARS-CoV-2 variants, nevertheless it would be necessary to verify its LoD for the new variants that will eventually emerge (page 22).

  1. Fig. 1: where are these mutations located? As shown by the authors, the in silico predictions do not totally correspond to the ‘real’ results (even with mutations, the assays work) – this is linked to the position of the mutation, and this information is not available and should be added. The method used by the authors for in silico analysis is based on number of mismatches only, without taking other factors into account (as was done in other studies with more complex tools used for in silico performance analysis). This limitation should be acknowledged in the discussion.

Authors: thank you for this criticism. We agreed with your point, and provide a new Figure 1 where some alignments between primers/probes and their consensus sequences are shown as examples and we provide all alignments in Supplementary materials (Supplementary data file I). This issue was also extensively discussed in the discussion section (see my previous answer to point 1).

  1. The LOD of the UCE assay was tested on the omicron synthetic genome, using the ‘optimized’ annealing temperature. How will this LOD ‘hold’ for the other current and future variants? This should be discussed.

Authors: Please see our answer to point 2.

  1. ‘Interestingly, two wastewater samples collected at Amsterdam Schiphol and Frankfurt Main airports tested positive for Omicron two weeks prior to the WHO officially designed Omicron a VOC, providing evidence that Omicron had already arrived in Europe before the WHO announcement (data not shown).’ Such important statement cannot be made without showing the data. So either these should be added, or the statement should be removed from results and discussion.

Authors: Thank you for your comment which gave us the possibility to further extend this section of the results. We decide to show these results in the revised version of our manuscript and we prepared a new figure illustrating them (Figure 6).

  1. “We tested the JRC-CoV-UCE method in comparison with the 2019-nCoV N2 and E Sarbeco methods on SARS-CoV-2 RNAs extracts …”. Why were these 2 assays selected for comparison? This just be clarified. This might also give some more information (which is needed) as to which extend the UCE assays are the ‘first’ universal assay or whether other assays could also be a universal one. This should be added to the discussion. Similar to this ‘….while the NSP9 region was already a target for the WHO recommended assay RdRp-nCoV-IP2 developed at the Institute Pasteur, France’. Why wasn’t then this assay chosen for comparison? Wouldn’t this also be a universal assay?

Authors: we selected the 2019-nCoV N2 and E Sarbeco assays as control because the two assays were considered by many laboratories as reference methods at the time in which the analyses were performed. Our results show that both methods. are still very reliable against Omicron.The JRC-CoV-UCE assay is, at the best of our knowledge, the only assay that was designed taking into consideration the ultra conserved elements of the SARS-CoV-2 RNA genome. We show that this assay has the lowest number of mismatches (higher degree of conservation) with all deposed GISAID SARS-CoV-2 sequences, while other WHO recommended assays, including the RdRp-nCoV-IP2 developed at the Institute Pasteur, France’ present more mismatches (Figure S1 and Figure 3).

  1. The discussion should reflect on the use of digital PCR compared to RT-qPCR for wastewater applications.

Authors: Following your suggestion, we have discussed the pros and cons of ddPCR in comparison to RT-qPCR for wastewater applications and provided relevant references on the topic (discussion page 18).

  1. The discussion is at some occasions a repetitions of the results. This should be carefully checked and modified (some suggestions have been made as to how the discussion should/could be extended and go beyond repeating the results).

Authors: The discussion section was extensively revised and extended taking into consideration the Reviewers’ comments and suggestions.

 Minor comments:

  • -Abstract: have these variants really been isolated? Seems rather difficult for a virus. They have been identified/sequenced. This sentence should be modified.

Authors: We agree and modified accordingly. Thank you.

  • Abstract: to indicate that environmental is waste water.

Authors: we changed environmental with wastewater.

  • Introduction: to add the state of the art of the performance evaluation of the WHO recommended assays, both in silico and in vitro. Other studies have already done similar investigations. These should be mentioned. In the discussion a part should be added how these relate to the current findings.

Authors: we inserted in the introduction a sentence mentioning other studies that evaluated before us the performance of the WHO recommended assays.

  • Fig 2. the legend and materials & methods should explain the ‘consensus seq’ versus the 'total seq’ results.

Authors: as mentioned above we extensively revised the Materials and Methods section. We hope that we have clarified the difference between consensus vs total sequences.

  • On which samples was the in silico PCR verified? Synthetic genomes?

Authors: we have clarified in the Materials and Methods section that the in silico PCR was performed using the SARS-CoV-2 consensus sequences. These sequences are representative of more than10 million GISAID high quality SARS-CoV-2 sequences.

  • Fig 3 right panel: why are there no dots for 2019-NCOV-N1?

Authors: the analysis was not performed for this method at this concentration of synthetic RNA. We have clarified this point in the new version of Figure 2 (ex Fig. 3).

  • Fig. 4: what is the SARS-cov-2 genome? The Wuhan genome? To what was the comparison done to identify the mutations?

Authors: Yes, the reference sequence was the Wuhan H1 genome. This information is now present in the Materials and Methods section.

  • Fig. 5: it is not clear form the figure if each variant was detected with both UCE assays (as it was a duplex).

Authors: In the new Figure 4 A (ex Fig. 5) we show the amplification plots of both UCE assays. These assays were always run together in duplex format.

  • Fig. 5C: why was the LOD performed for the single assays and not for the duplex?

Authors: The assays were always run together in a duplex format. We apologize that this was not so clear in the previous version of the manuscript. We moved Fig. 5 C to Supplementary materials (Figure S4) as we provide a new figure for the LoD calculation (Figure 4D).

  • ‘Based on these results, in accordance with WHO guidelines, we would also like to recommend to use diagnostic and wastewater tests that include multiple gene targets rather than single target assays to reduce at the minimum the risk that some SARS-CoV-2 would escape detection.’ The authors are not the first to recommended this. This should be made clear (reformulation the sentence + adding appropriate references).

Authors: With reference to your comment, we have revised this sentence accordingly (discussion page 17).

  • ‘It is possible that SARS-CoV-2 will acquire further genomic changes falling into the sequences recognized by primers and probes of these assays and therefore reducing their diagnostic accuracy. An example is the variant of interest BA.2.75, which shows 75 base changes and four deletions of nine bps or more.’ Was this tested (in silico or in real) by the authors? Can this be added?

Authors: The clinical samples used did not include BA.2.75 sublineage of the omicron variant as there were insufficient numbers of available samples in the LuxMicroBiobank at the time of sample requests for the experiments throughout this manuscript. However, we performed in silico alignment considering all Omicron sub-lineages including the more recent BA.275, BQ.1 and XBB (Figure S2). We show that the number of mismatches between the oligonucleotides of the WHO recommended assays and these sequences did not increase, suggesting that most likely these assays still will detect with similar efficiency these more recent sub-linages. Similarly, also the JRC-CoV-2 assay did not present mismatches, that may predict a drop in performance, with these more recent sub lineages. This observation was mentioned in the discussion section.

  • ‘The in silico PCR simulation approach used in this study, accurately predicted the performance’. This is not totally correct, as the ‘real’ results were different from the in silico results for some assays (better results obtained with ‘real’ assay, compared to in silico. See also comment above on this. The sentence should be reformulated.

Authors: We agree with the Reviewer. The sentence was reformulated accordingly.

  • ‘SARS-CoV-2 has acquired a large number of mutations throughout its genome.’ The Wuhan strain?

Authors: Yes, it is. Specified in the text.